# Endogenous Fructose Production and Metabolism Drive Metabolic Dysregulation and Liver Disease in Mice with Hereditary Fructose Intolerance

**DOI:** 10.3390/nu15204376

**Published:** 2023-10-16

**Authors:** Ana Andres-Hernando, David J. Orlicky, Masanari Kuwabara, Christina Cicerchi, Michelle Pedler, Mark J. Petrash, Richard J. Johnson, Dean R. Tolan, Miguel A. Lanaspa

**Affiliations:** 1Division of Endocrinology, Metabolism and Diabetes, University of Colorado Denver, Aurora, CO 80045, USA; ana.andreshernando@cuanschutz.edu; 2Department of Pathology, University of Colorado School of Medicine, Aurora, CO 80045, USA; david.orlicky@cuanschutz.edu; 3Department of Cardiology, Toranomon Hospital, Tokyo 105-8470, Japan; kuwamasa728@gmail.com; 4Division of Public Health, Center for Community Medicine, Jichi Medical University, Tochigi 329-0431, Japan; 5Division of Renal Diseases and Hypertension, University of Colorado Denver, Aurora, CO 80045, USA; ccicerchi1@yahoo.com (C.C.); richard.johnson@cuanschutz.edu (R.J.J.); 6Department of Ophthalmology, University of Colorado School of Medicine, Aurora, CO 80045, USA; michelle.pedler@cuanschutz.edu (M.P.); mark.petrash@cuanschutz.edu (M.J.P.); 7Department of Biology, Boston University, Boston, MA 02215, USA; tolan@bu.edu

**Keywords:** HFI, endogenous fructose, aldolase b, polyol pathway, metabolic dysfunction

## Abstract

Excessive intake of sugar, and particularly fructose, is closely associated with the development and progression of metabolic syndrome in humans and animal models. However, genetic disorders in fructose metabolism have very different consequences. While the deficiency of fructokinase, the first enzyme involved in fructose metabolism, is benign and somewhat desirable, missense mutations in the second enzyme, aldolase B, causes a very dramatic and sometimes lethal condition known as hereditary fructose intolerance (HFI). To date, there is no cure for HFI, and treatment is limited to avoiding fructose and sugar. Because of this, for subjects with HFI, glucose is their sole source of carbohydrates in the diet. However, clinical symptoms still occur, suggesting that either low amounts of fructose are still being consumed or, alternatively, fructose is being produced endogenously in the body. Here, we demonstrate that as a consequence of consuming high glycemic foods, the polyol pathway, a metabolic route in which fructose is produced from glucose, is activated, triggering a deleterious mechanism whereby glucose, sorbitol and alcohol induce severe liver disease and growth retardation in aldolase B knockout mice. We show that generically and pharmacologically blocking this pathway significantly improves metabolic dysfunction and thriving and increases the tolerance of aldolase B knockout mice to dietary triggers of endogenous fructose production.

## 1. Introduction

Hereditary fructose intolerance (HFI) is an autosomal recessive disorder due to mutations in the aldolase B gene, and has a frequency of about 1:10,000 in the general population [1,2]. Aldolase B is the second enzyme in fructose metabolism and converts fructose-1-phosphate (F1P) into glyceraldehyde and dihydroxyacetone phosphate. When aldolase B is mutated, F1P accumulates, leading to low intracellular phosphate and energy imbalance. Further, F1P stimulates rapid glucose uptake by releasing glucokinase regulatory protein (GKRP), causing acute hypoglycemia, intracellular ATP depletion and uric acid generation. Experimental studies suggest that all manifestations of the disease can be blocked by inhibiting fructokinase, the first enzyme in fructose metabolism, as this blocks both the accumulation of fructose-1-phosphate and the adenine nucleotide turnover [3].

Since fructokinase inhibitors are not available, the classical management is to simply reduce all exposure to fructose, which requires avoidance of foods with added sugars (such as sucrose or high fructose corn syrup), as well as natural foods that may contain fructose (such as fruits and honey). This can be difficult in western society, where fructose is contained in up to 70% of processed foods. As such, subjects with HFI can develop chronic manifestations of disease, most notably non-alcoholic steatohepatitis (NASH), elevated blood pressure and Fanconi syndrome [1,4]. 

The assumption in HFI management is that the treatment objective is to reduce dietary fructose. However, recently, it has been shown that fructose can also be produced endogenously via the conversion of glucose to sorbitol and then fructose via the aldose reductase-sorbitol dehydrogenase (polyol) pathway [5,6,7]. This latter pathway can be activated by a variety of mechanisms, including a high glucose diet, high serum osmolality induced by salt or alcohol, and even by hyperuricemia itself [5,6,7,8,9]. Thus, it is possible that the chronic liver disease that develops commonly in subjects with HFI might be partially due to endogenous fructose exposure, particularly when even traces of fructose can trigger the disease [4]. Indeed, some studies suggest that the liver disease in HFI can occur independently of reducing dietary fructose [4].

Therefore, we decided to test the hypothesis that endogenous fructose production might have a role in driving disease manifestations in a model of HFI in mice.

## 2. Materials and Methods

Study Approval: All animal experiments were conducted with adherence to the National Institutes of Health Guide for the Care and Use of Laboratory Animals. The animal protocol was approved by the Institutional Animal Care and Use Committee of the University of Colorado (Aurora, CO, USA).

Animal Experiments: Aldolase B KO mice in a C57BL/6J genetic background were generated and obtained from Dr Dean Tolan at Boston University and had been previously characterized [3,10]. Ketohexokinase (fructokinase, Khk) KO (B6;129-Khktm2Dtb) mice were originally developed by David Bonthorn at Leeds University (UK) [11] and were bred and maintained at the Univ. Colorado with pure C57/Bl6 for over 7 generations to ensure the mice were on the B6 genetic background. On receipt, mice were bred with pure C57BL/6J mice to obtain both knockout and WT littermates and further crossed with aldolase B KO mice to obtain aldolase B/Khk double KO mice. All experimental mice were maintained in temperature- and humidity-controlled specific pathogen-free conditions on a 14 h dark/10 h light cycle and allowed ad libitum access to a fructose-free diet (Bioserv, catalog F6700). Food and water (plain or containing glucose, sorbinil, sorbitol or ethanol) were measured daily. All experiments were conducted with adherence to the National Institutes of Health Guide for the Care and Use of Laboratory Animals. In all experiments, 8-week-old male mice were used. An a priori power analysis was not performed for determining sample size, but instead, prior experience with the model was used to establish a sample size sufficient to delineate therapeutic efficacy. Mice were randomly assigned to the dietary regimens using the Random function in Excel, and all data were analyzed in a blinded fashion. No mice were excluded from the study due to health concerns or any other criterion.

Histopathology: Formalin-fixed paraffin-embedded liver and kidney sections were stained with H&E (for lipid and inflammation), periodic acid–Schiff (PAS, for glycogen), or Picro-Sirius Red (PSR, for fibrosis). Histological examination was performed as previously described [3]. For liver injury scoring, the entire cross section of liver including all zones was analyzed from each mouse. Images were captured on an Olympus BX51 microscope equipped with a 4 megapixel Macrofire digital camera (Optronics) using the PictureFrame Application 2.3 (Optronics). Composite images were assembled with the use of Adobe Photoshop. All images in each composite were handled identically. To quantitate the fibrosis, 10 polarized images were made in a tiling fashion across each PSR-stained slide, then quantitated and averaged using the 3I Slidebook program to arrive at the PSR-stained pixels per 100× field for that slide/animal. The criteria used in liver injury scoring were modified from the criteria used by Brunt [12]. The criteria used included: (a) liver cell injury (ballooning, acidophil bodies, necrotic cells, pigmented macrophages, megamitochondria, etc.); (b) inflammation (lobular inflammation, foci of inflammatory cells, lipogranulomas, portal inflammation, Langhans giant cells, etc.); (c) steatosis (macro and microvesicular steatosis); (d) fibrosis (perisinusoidal, periportal, bridging fibrosis, cirrhosis, etc.); and (e) other features (glycogenated nuclei, mitotic figures, etc.). All animals were scored for all criteria, then scores for each animal for each criterion were averaged to determine a mean ± SD for each group.

Western blotting: Protein lysates were prepared from mouse tissue employing lysis buffer containing 0.5% triton X-100, 50 mM β-glycerophosphate, 2 mM MgCl2, 1 mM EGTA, 1 mM dithiothreitol and a cocktail of protease inhibitors (Roche, Minato City, Tokyo). Protein content was determined by the bicinchoninic acid (BCA) protein assay (Pierce). Total protein (50 μg) was separated by SDS-PAGE (10% *w*/*v*), and transferred to PVDF membranes (BioRad, Hercules, CA, USA). Membranes were first blocked for 1 h at 25 °C in 4% (*w*/*v*) instant milk dissolved in 0.1% Tween-20 Tris-buffered saline (TTBS), and incubated with the following primary rabbit or mouse-raised antibodies (1:1000 dilution in TTBS): aldolase B (Abnova, Taipei City, Taiwan, H00000229-A01), GCK (Abnova, H000026465-B02P), KHK (Sigma, St. Louis, MI, USA, HPA007040), and SDH (Proteintech, Koto City, Tokyo, 15881). The antibody to AR was custom-made in rabbit and previously validated [6]. Membranes were visualized using an anti–rabbit (7074) or anti–mouse IgG (7076) horseradish peroxidase–conjugated secondary antibody (1:2000, Cell Signaling, Danvers, MA, USA) using the HRP Immunstar detection kit (Bio-Rad, Hercules, CA, USA). Chemiluminescence was recorded with an Image Station 440CF, and the results were analyzed with 1D Image Software 3.6 (Kodak Digital Science, Rochester, NY, USA).

Biochemical and Tissue Analysis: Blood was collected in Microtainer tubes (BD) from cardiac puncture of mice under isoflurane, and serum was obtained after centrifugation at 13,000 rpm for 2 min at room temperature. Determination of serum and urine parameters was performed biochemically following the manufacturer’s instructions: uric acid: DIUA-250, Phosphate: DIPI-500, ALT: EALT-100, AST: EAST-100 (Bioassay Systems Hayward, Hayward, CA, USA), Creatinine: C753291, Pointe Scientific, FGF21 (R&D Systems, Minneapolis, MN, USA, MF2100). Fractional excretion of phosphate and uric acid was calculated by using the following ratio: FEx = (UX ∗ PCr ∗ 100)/(Px ∗ UCr), where U and P refer to the urine and plasma concentrations of phosphate, uric acid and creatinine (Cr). Determination of parameters in tissue was performed in freeze-clamped tissues and measured following the manufacturer’s protocols (Uric acid DIUA-250, Sorbitol ESBT-100 and fructose EFRU-100, all from Bioassay Systems). To extract metabolites from liver samples, frozen liver samples were ground at liquid nitrogen temperature with a Cryomill (Retsch). The resulting tissue powder was weighed (~20 mg). The extraction was then performed by adding −20 °C extraction solvent to the powder and incubating it at −20 °C overnight, followed by vortexing and centrifugation at 16,000× *g* for 10 min at 4 °C. The volume of the extraction solution (μL) was 40 times the weight of tissue (mg) to make an extract of 25 mg tissue per mlilliliter of solvent. Dried extracts were then redissolved in LC-MS Grade water (catalog 51140; Thermo Fisher, Waltham, MA, USA). Metabolites were analyzed via reverse-phase ion-pairing chromatography coupled to an Exactive Orbitrap mass spectrometer (Thermo Fisher Scientific). The mass spectrometer was operated in negative-ion mode with a resolving power of 100,000 at a mass-to-charge ratio (*m*/*z*) of 200 and a scan range of 75–1000 *m*/*z*. Energy charge was calculated as ([ATP] + 1/2[ADP])/[ATP] + [ADP] + [AMP] [13,14].

Statistical Analysis: All numeric data are presented as means  ±  SE. Independent replicates for each data point (*n*) are shown in the figures. Data graphics and statistical analysis were performed using Prism 5 (GraphPad). Data without indications were analyzed by one-way ANOVA with a Tukey post hoc test. *p* values of <0.05 were regarded as statistically significant.

## 3. Results

Effects of glucose-rich solutions on metabolic dysregulation in aldolase B KO mice. The activation of the polyol pathway is initiated by the up-regulation of aldose reductase (AR), which in the liver results in the production of sorbitol from glucose; that sorbitol is then metabolized to fructose by sorbitol dehydrogenase (SDH, Figure 1). Fructokinase (ketohexokinase, KHK) then phosphorylates fructose to Fructose-1-phosphate (F1P) as the first step of fructolysis. However, in aldolase B deficient mice, F1P is not further metabolized and thus accumulates, causing phosphate sequestration, low energy charge, energy imbalance and metabolic dysregulation ([3]). Of interest, and as shown in Figure 2A, exposure of mice to varying concentrations of glucose (from 0 to 15%) significantly diminishes growth in aldolase B KO mice in a dose-dependent manner. Specifically, we observed that solutions greater than 5% glucose, which are similar to those found in regular sweet beverages, were associated with a marked failure to thrive (28.3 ± 5.6% increase in body weight gain in non exposed aldolase B KO mice versus −5.2 ± 2.2% decrease in body weight in mice consuming a 15% glucose solution, *p* < 0.01) and significantly higher levels of F1P in their livers (Figure 2B). The effect of glucose solutions on growth retardation and liver disease in aldolase B KO mice was drastically different from that observed in wild-type mice exposed to the same amounts of glucose (Figure 2C,D) and would point to a strain-specific effect derived from glucose metabolism in mice deficient in aldolase B. Similarly to our previous reports on the induction of the polyol pathway by glucose-rich solutions [6], glucose up-regulated AR expression (Figure 2E) and activated the polyol pathway characterized by high sorbitol and fructose levels in the gut and liver of aldolase B KO mice (Figure 2F,G). Thus. to determine the importance of AR and the polyol pathway in glucose-dependent growth retardation in aldolase B KO mice, we decided to test the efficiency of sorbinil, a well-characterized AR inhibitor [15,16]. To this end, we first exposed aldolase B KO mice to 0% control or 15% glucose solutions for 6 weeks, and then animals were subsequently divided into the same glucose solutions containing either vehicle or sorbinil (0.25 mg/mL) for another 6 weeks (Figure 2H, left). Of interest, administration of sorbinil was associated with a significant increase in body weight gain in 15% glucose-fed mice compared to vehicle-receiving mice (9.6 ± 2.6% increase in body weight gain in sorbinil-treated aldolase B KO mice receiving 15% glucose versus 4.1 ± 1.2% decrease in body weight in aldolase B KO mice receiving vehicle, *p* < 0.01, Figure 2H right). The increase in growth by sorbinil in glucose-exposed aldolase B KO mice was associated not only with significantly lower hepatic levels of sorbitol and fructose (Figure 2I,J) reflecting decreased activity through the polyol pathway, but also with lower levels of F1P (Figure 2K) and a marked improvement in overall energy charge (Figure 2L). Furthermore, the blockade of AR in aldolase B KO mice was associated with a marked improvement of liver injury, as denoted by fat mobilization and low liver steatosis and inflammation (Figure 2M), injury score (Figure 2N) and significantly lower plasma levels of liver transaminases ALT and AST (Figure 2O,P). Together, our data suggest that glucose maximizes the deleterious metabolic consequences of the activation of the polyol pathway in aldolase B KO mice.

Sorbitol exacerbates metabolic dysregulation in aldolase B KO mice. In foods, sorbitol, an intermediate product of the polyol pathway, is often considered a safer alternative to caloric sugars, as it provides less calories per gram than table sugar and it is not fully digested in the small intestine. However, and consistent with our hypothesis of the key importance of endogenous fructose in HFI, exposure of aldolase B KO mice to sorbitol-containing solutions dramatically impaired thriving in these mice (Figure 3A). Specifically, solutions containing as little as 1 or 5% sorbitol resulted in a reduction of body weight of 4.5 ± 1.8 and 15.4 ± 4.6%, respectively (*p* < 0.01). As with glucose, the effect of sorbitol in growth retardation was not observed in wild-type mice receiving the same amounts of sorbitol (Figure 3B) and would point to the importance of the polyol pathway in the pathogenesis of HFI. This is consistent with intrahepatic levels of fructose and F1P being significantly higher in aldolase B KO mice receiving sorbitol compared to water control (Figure 3C,D). Furthermore, and compared to glucose, sorbitol administration produced a much greater exacerbation of the pathogenesis of HFI, as denoted by greater F1P accumulation (Figure 3D), and liver injury score (Figure 3E), liver steatosis, inflammation (Figure 3F) and fibrosis (Figure 3G). The observed failure to thrive induced by glucose and sorbitol can be the consequence of the overall low energy state, as shown in Figure 2L, as well as the metabolic consequences associated with Fanconi syndrome, a defect associated with growth retardation and energy wasting that is common in subjects with HFI and the result of renal tubular acidosis and urinary glucose turnover [3,17,18,19,20,21]. Consistent with renal consequences of sorbitol in HFI, levels of fructose in the kidney cortex were upregulated in sorbitol-exposed aldolase B KO mice (58.1 ± 29.3 nmol/mg versus 1.4 ± 1.0 nmol/mg, *p* < 0.01, Figure 3H), suggestive of the activation of the polyol pathway in renal tubules. Furthermore, and even though we did not observe significant pathology in the kidney cortex of sorbitol-exposed aldolase B KO mice (Figure 3I), mice demonstrated features associated with Fanconi syndrome and renal acidosis, including higher fractional excretion of uric acid and phosphate (Figure 3J,K), and more importantly, substantial glucosuria (Figure 3L), which would help explain the growth retardation in these mice.

Blockade of KHK protects against glucose and sorbitol-induced metabolic dysregulation in aldolase B KO mice. The activation of the polyol pathway promotes multiple metabolic effects that could participate in the pathogenesis of HFI. These include NADPH depletion, hyperproduction of NADH, and redox imbalances or high production of osmolytes like sorbitol [22,23]. We have previously shown that blocking KHK markedly improves HFI in fructose-fed aldolase B KO mice [3]. Therefore, we hypothesized that of all of the metabolic consequences associated with the activation of the polyol pathway, the blockade of the metabolism of endogenous fructose would be particularly important against the deleterious effects of glucose and sorbitol in aldolase B KO mice (Figure 4A). To this end, we tested the response of aldolase B and Khk double KO mice (AldoB/Khk DKO, Figure 4B) to glucose and sorbitol solutions and compared their response to that of regular aldolase B KO mice. First, and as shown in Figure 4C,D, we found that AldoB/Khk DKO mice consumed significantly more glucose and sorbitol solutions than aldolase B KO mice, indicating that the deletion of Khk is associated with much more tolerance to intermediates of the polyol pathway by aldolase B KO mice. Consistently, AldoB/Khk DKO mice on glucose (Figure 4E) and sorbitol (Figure 4F) demonstrated a marked increase in body weight gain compared to aldolase B counterparts. Greater body weight gain was associated with a marked improvement in metabolic dysregulation in the liver, denoted by the reduction in hepatic F1P levels (Figure 4G) and greater energy charge (Figure 4H) with reduced liver injury (Figure 4I), steatosis, inflammation (Figure 4J), fibrosis (Figure 4K) and liver dysfunction (Figure 4L,M). Further, levels of FGF21, a hepatokine produced to ameliorate liver injury and reduce sugar intake, were remarkably high in aldolase B knockout mice in response to glucose and sorbitol and blunted in AldoB/Khk DKO mice, suggestive of improved liver injury when KHK expression was deleted (Figure 3N,O).

Together, our data suggest that the deleterious effects of glucose and sorbitol in aldolase B KO mice are driven by the production and metabolism of fructose.

Deleterious mechanistic effects of ethanol in HFI via the polyol pathway. The up-regulation of aldose reductase and the activation of the polyol pathway can be promoted by factors other than high glucose. These factors include high osmolality [24], hypoxia [25,26,27] and uric acid [8,9], which are associated with chronic ethanol exposure [9,28,29]. Based on this, we hypothesized that ethanol promotes the activation of the polyol pathway and the production of endogenous fructose, whose metabolism via KHK would be particularly deleterious in aldolase B KO mice (Figure 5A) due to the accumulation of F1P. Consistently, plasma osmolality in aldolase B KO mice receiving a 2.5 g/kg ethanol dose via oral gavage resulted in a significant up-regulation of osmolality in the portal vein up to 365 ± 12 mOsm/KgH_2_O after 10 min (*p* < 0.01, Figure 5B). Further, ethanol-fed wild-type and aldolase B KO mice demonstrated a marked up-regulation of liver AR and KHK expression (Figure 5C) in parallel with significantly higher levels of sorbitol (6.63-fold, *p* < 0.01 Figure 5D), fructose (16.2-fold, *p* < 0.01 Figure 5E) and F1P (3.1-fold, *p* < 0.01 Figure 5F), indicating that ethanol efficiently activates the polyol pathway and promotes endogenous fructose metabolism in the liver of mice. To determine the importance of endogenous fructose metabolism in alcoholic liver disease in aldolase B KO mice, we subsequently analyzed the response to ethanol of control aldolase B KO and AldoB/Khk DKO mice. As shown in Figure 5G, deletion of Khk in aldolase B KO mice significantly reduced F1P levels in response to ethanol (90.5 ± 3.5% reduction, *p* < 0.01). Further, AldoB/Khk DKO mice demonstrated greater preference for ethanol in two-bottle choice paradigms (Figure 5H) and consumed significantly more ethanol in single-bottle (6% *w*/*v*) (Figure 5I) than aldolase B KO counterparts, indicative of a much greater tolerance to ethanol of KHK deficient aldolase B KO mice. Consistently, and unlike aldolase B KO mice, which failed to thrive on ethanol, AldoB/Khk DKO mice had greater body weight gain (Figure 5J) with improved liver injury, as denoted by reduced inflammation, steatosis (Figure 5K) and overall, a significantly lower injury score (Figure 5L) and levels of transaminases ALT and AST in plasma (Figure 5M,N).

Collectively, we present compelling evidence that the activation of the polyol pathway and the production of endogenous fructose elicit important deleterious effects in mice deficient for aldolase B and potentially could be clinically relevant in people with HFI, as they are highly sensitive to fructose. The activation of the polyol pathway can be triggered by different stimuli, including glucose-containing foods, sorbitol or ethanol. In consequence, the metabolism of endogenous fructose by KHK results in a marked accumulation of F1P, causing metabolic dysregulation, liver dysfunction and injury and failure to thrive.

## 4. Discussion

Hereditary fructose intolerance is an orphan disease with dramatic and potentially lethal consequences. One of the key aspects of this condition is that there is no cure, and treatment is often limited to just avoiding foods containing fructose. This is a very challenging approach, as fructose, either as table sugar or high fructose corn syrup, is a common additive in foods to stimulate appetite and palatability. Besides, even though subjects with HFI “learn” to avoid dietary fructose, individuals with this condition often manifest a clinically ill condition [30]. This observation would suggest that either the avoidance of dietary fructose is not fully achieved or, alternatively, fructose is being endogenously produced and metabolized in the body as a consequence of their dietary behavior.

Dietary guidelines for HFI recommend the substitution of fruits, breads, desserts or sodas by vegetables and grains that often contain a high glycemic index like rice, potatoes or pasta [31]. In this regard, we previously showed that foods with a high glycemic index stimulate the specific activation of the polyol pathway in the liver and that the metabolism via KHK of the fructose endogenously produced is an important deleterious step in the pathogenesis of metabolic syndrome, diabetes and its complications [5,6]. Thus, our study builds upon our previous reports on the importance of the polyol pathway in metabolic dysfunction and further emphasizes a key role for endogenous fructose in a model of metabolic dysregulation that maximizes the deleterious effects of this sugar. In these settings, our work demonstrates that people with HFI will be more sensitive to the deleterious effects associated with the activation of the polyol pathway and, therefore, guidelines aimed at monitoring and controlling the intake of glycemic-rich foods as well as external activators of the pathway like alcohol should be promoted. The activation of the polyol pathway was observed in those tissues that preferentially metabolize dietary fructose, the small intestine and the liver [32,33]. Even though our study was mainly focused on the consequences of endogenous fructose metabolism in the liver, it is important to acknowledge the potential relevance of the activation of the polyol pathway in the gut. In this regard, as it is known that dietary fructose causes leaky gut and portal endotoxemia [34], it is likely that intestinal metabolism of endogenous fructose could potentiate liver disease and HFI in aldolase B knockout mice. Therefore, studies aimed at parsing out the specific roles of intestinal and hepatic activation of AR in HFI are warranted.

The mechanisms whereby the accumulation of F1P in the hepatocyte causes massive metabolic dysregulation are multiple. On one hand, F1P itself has glycolytic properties, as it stimulates the dissociation between glucokinase and its regulatory protein in the nucleus. The disruption of glucokinase with its regulatory protein promotes its translocation to the cytosol and a rapid glycolytic influx with subsequent hypoglycemia and risk for hypoglycemic shock [3]. The hypoglycemia is also maintained by a marked glycogen storage disorder type VI in which glycogen synthesis is hyper-stimulated with minimal glycogenolysis [35]. Further, the sequestration of phosphate in F1P impairs proper energy balance and promotes a generalized low energy charge in the liver, driving inflammation. This is maximized by a lack of activation of the energy sensor 5′ AMP-activated kinase (AMPK) in HFI [3]. Reduced AMPK-specific phosphorylation and inhibition of glycogen synthase and acetyl-CoA Carboxylase results in further glycogen accumulation and liver steatosis. The combination of steatosis plus the inflammation associated with a low energy state then promotes fibrosis and the transition to non-alcoholic steatohepatitis (NASH) often observed in subjects with HFI. The reduced appetite for sugar and alcohol in aldolase B knockout mice could also be dependent on central mechanisms. In this regard, [5] FGF21 is a liver-derived hormone that, in response to sugar and alcohol intake, is secreted by hepatocytes and acts on the hypothalamus to reduce their intake [36,37,38,39]. Our data demonstrate that aldolase B knockout mice on glucose and sorbitol respond to these sugars by producing much higher amounts of FGF21 than wild-type mice. The high FGF21 in aldolase B knockout mice could be a response to minimize intake of glucose and sorbitol centrally in response to a much-exacerbated liver injury in these mice. Consistently, FGF21 levels drop when Khk expression is deleted, similarly to what we observed in sugar-exposed liver-specific Khk knockout mice [34], suggesting that the benefits of knocking out KHK outweigh the detrimental effect that glucose and sorbitol exert on aldolase B knockout mice, and thus, the up-regulation of FGF21 is blunted.

Besides NASH, our study also points to other glucose-independent factors promoting liver disease in HFI. These factors would independently trigger the activation of the polyol pathway and include ethanol. Our study identifies a new mechanism, alcohol consumption, predisposing people with aldolase B mutations to the deleterious effects of HFI. Even though alcoholic and non-alcoholic liver disease are two different entities etiologically, our study would suggest that they both share a mechanistic common link, the activation and promotion of fructose metabolism. A role for sugar, and specifically fructose, in the development and progression of fatty liver disease has been known for decades [32,40]. We present here evidence that in aldolase B KO mice, the metabolism of non-dietary fructose via KHK could represent a novel and important step in the pathogenesis of alcoholic liver disease as well.

In summary, we propose that the specific activation of endogenous fructose production and metabolism in the liver could be a potential driver of the disease in humans, as these subjects are sensitive to even traces of fructose [3]. Therefore, we further propose that controlling the intake of glucose, and particularly foods with a high glycemic index, as well as alcohol could be clinically relevant to avoid the chronic consequences associated with HFI.

## Figures and Tables

**Figure 1 nutrients-15-04376-f001:**
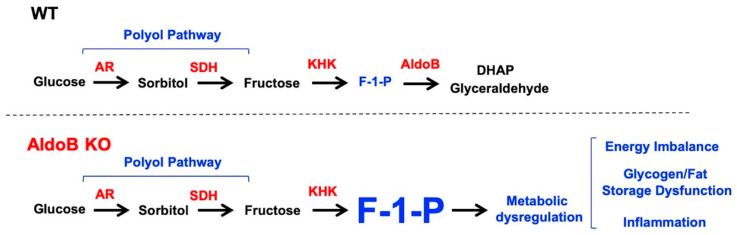
Proposed schematic of the relevance of the activation of the polyol pathway in aldolase B knockout mice and its potential implications for HFI. The up-regulation of aldose reductase (AR) produces sorbitol and endogenous fructose. While in wild-type (WT) mice, endogenous fructose is metabolized by KHK and aldolase b, limiting the accumulation of fructose-1-phosphate (F1P), in aldolase B KO mice, F1P is accumulated, leading to metabolic dysregulation characterized by energy imbalance, demobilization of fat and glycogen stores and inflammation.

**Figure 2 nutrients-15-04376-f002:**
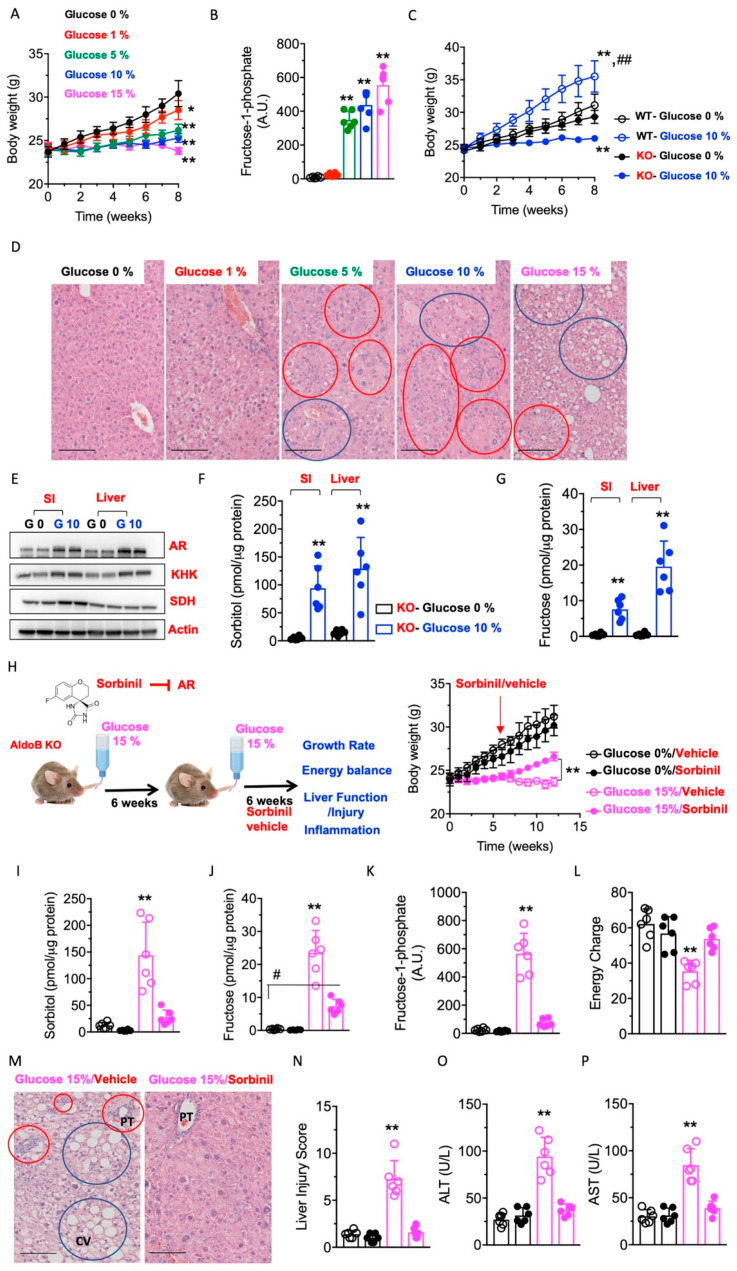
Activation of the polyol pathway and hepatic metabolic dysregulation in aldolase B KO mice. (**A**) Body weight gain in aldolase B KO mice consuming different concentrations of glucose solution (from 0 to 15% *w*/*v*) for 8 weeks. (**B**) Intrahepatic F1P levels in the same mice as in (**A**). (**C**) Comparison of body weight gain in wild-type (WT) and aldolase B KO mice consuming 0 or 10% glucose *w*/*v* solutions for 8 weeks. (**D**) Representative liver H&E images from aldolase B KO mice consuming increasing amounts of glucose. Blue circles denote macrosteatotic areas, and red circles indicate areas with ductal reaction and inflammation. Size Bar: 50 µM. (**E**) Representative Western blot from small intestine and liver extracts for AR and actin control in aldolase B KO mice on 0 or 10% glucose for 8 weeks. (**F**) SI and liver sorbitol levels in the same mice as in (**E**). (**G**) SI and liver fructose levels in the same mice as in (**E**). (**H**) Left, schematic depicting the interventional approach with sorbinil (0.25 mg/mL) in glucose-fed mice. Right, body weight gain in aldolase B KO mice consuming 0 (black symbols) or 15% (pink symbols) glucose solutions for 12 weeks and receiving vehicle or sorbinil from week 6. (**I**) Intrahepatic sorbitol levels in the same mice as in (**H**). (**J**) Intrahepatic fructose levels in the same mice as in (**H**). (**K**) Intrahepatic F1P levels in the same mice as in (**H**). (**L**) Intrahepatic energy charge in the same mice as in (**H**). (**M**) Representative liver H&E images from aldolase B KO mice consuming 15% glucose alone or in combination with sorbinil as in (**H**). Blue circles denote macrosteatotic areas, and red circles indicate areas with ductal reaction and inflammation. Size Bar: 50 µM. PT: Portal triad, CV: Central vein. (**N**) Liver injury score from the same mice as in (**G**). (**O**,**P**) Plasma ALT and AST levels in the same mice as in (H). The data in (**A**–**C**), (**E**–**L**) and (**N**–**P**) were presented as the means ± SEM and analyzed by two-tail *t*-test (**E**–**G**) and one-way ANOVA with Tukey post hoc analysis. For (**A**,**B**), * *p* < 0.05 and ** *p* < 0.01 versus 0% glucose. For (**C**), ** *p* < 0.01 versus respective WT or KO 0% glucose, ## ** *p* < 0.01 between WT-10% glucose and KO-10% glucose. For (**I**–**P**), ** *p* < 0.01 versus rest of the groups, # *p* < 0.05. *n* = 6 mice per group.

**Figure 3 nutrients-15-04376-f003:**
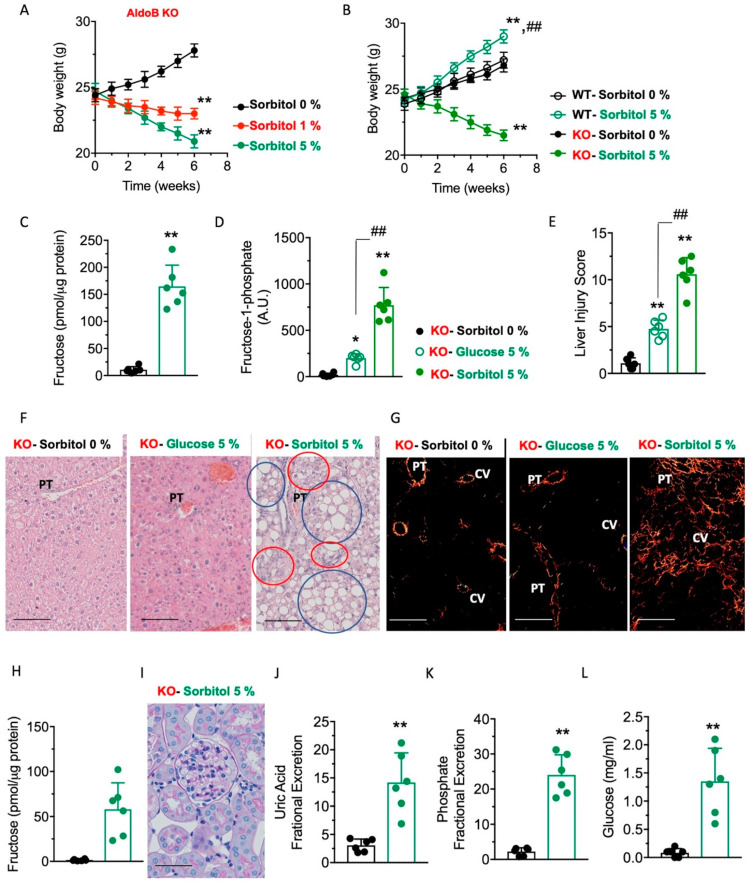
Effects of sorbitol supplementation on hepatic metabolic dysregulation in aldolase B KO mice. (**A**) Body weight gain in aldolase B KO mice consuming different concentrations of sorbitol (from 0 to 5% *w*/*v*) for 6 weeks. (**B**) Comparison of body weight gain in wild-type (WT) and aldolase B KO mice consuming 0 or 5% sorbitol (*w*/*v*) for 6 weeks. (**C**) Intrahepatic fructose levels in aldolase B KO mice consuming 0 or 5% sorbitol (*w*/*v*) for 6 weeks. (**D**) Intrahepatic F1P levels in aldolase B KO mice consuming 0 or 5% sorbitol (*w*/*v*) or glucose (*w*/*v*) for 6 weeks. (**E**) Liver injury score in aldolase B KO mice consuming 0 or 5% sorbitol (*w*/*v*) or 5% glucose (*w/v*) for 6 weeks. (**F**) Representative liver H&E images from aldolase B KO mice consuming 0 or 5% sorbitol (*w*/*v*) or 5% glucose (*w/v*) for 6 weeks. Red circles denote macrosteatotic areas, and blue circles indicate areas with ductal reaction and inflammation. Size Bar: 20 µM. PT: Portal triad. (**G**) Representative picro sirius red images under polarized light from aldolase B KO mice consuming 0 or 5% sorbitol (*w*/*v*) or 5% glucose (*w/v*) for 6 weeks. Size Bar: 20 µM. PT: Portal triad. CV: Central vein. (**H**) Intrarenal fructose levels in aldolase B KO mice consuming 0 or 5% sorbitol (*w*/*v*) for 6 weeks. (**I**) Representative kidney PAS images from aldolase B KO mice on 5% sorbitol for 6 weeks. (**J**) Urinary uric acid fractional excretion in aldolase B KO mice consuming 0 or 5% sorbitol (*w*/*v*) for 6 weeks. (**K**) Urinary phosphate fractional excretion in aldolase B KO mice consuming 0 or 5% sorbitol (*w*/*v*) for 6 weeks. (**L**) Urinary glucose excretion in aldolase B KO mice consuming 0 or 5% sorbitol (*w*/*v*) for 6 weeks. The data in (**A**–**E**), and (**J**–**L**) were presented as the means ± SEM and analyzed by two-tail *t*-test (**C**,**H**–**L**) and one-way ANOVA with Tukey post hoc analysis (**A**,**B**,**D**,**E**). For (**A**), ** *p* < 0.01 versus 0% sorbitol. For (**B**), ** *p* < 0.01 versus respective WT or KO 0% sorbitol and ## ** *p* < 0.01 between WT-5% sorbitol and KO-5% sorbitol. For (**D**,**E**), * *p* < 0.05 and ** *p* < 0.01 versus 0% sorbitol, ## *p* < 0.01. *n* = 6 mice per group.

**Figure 4 nutrients-15-04376-f004:**
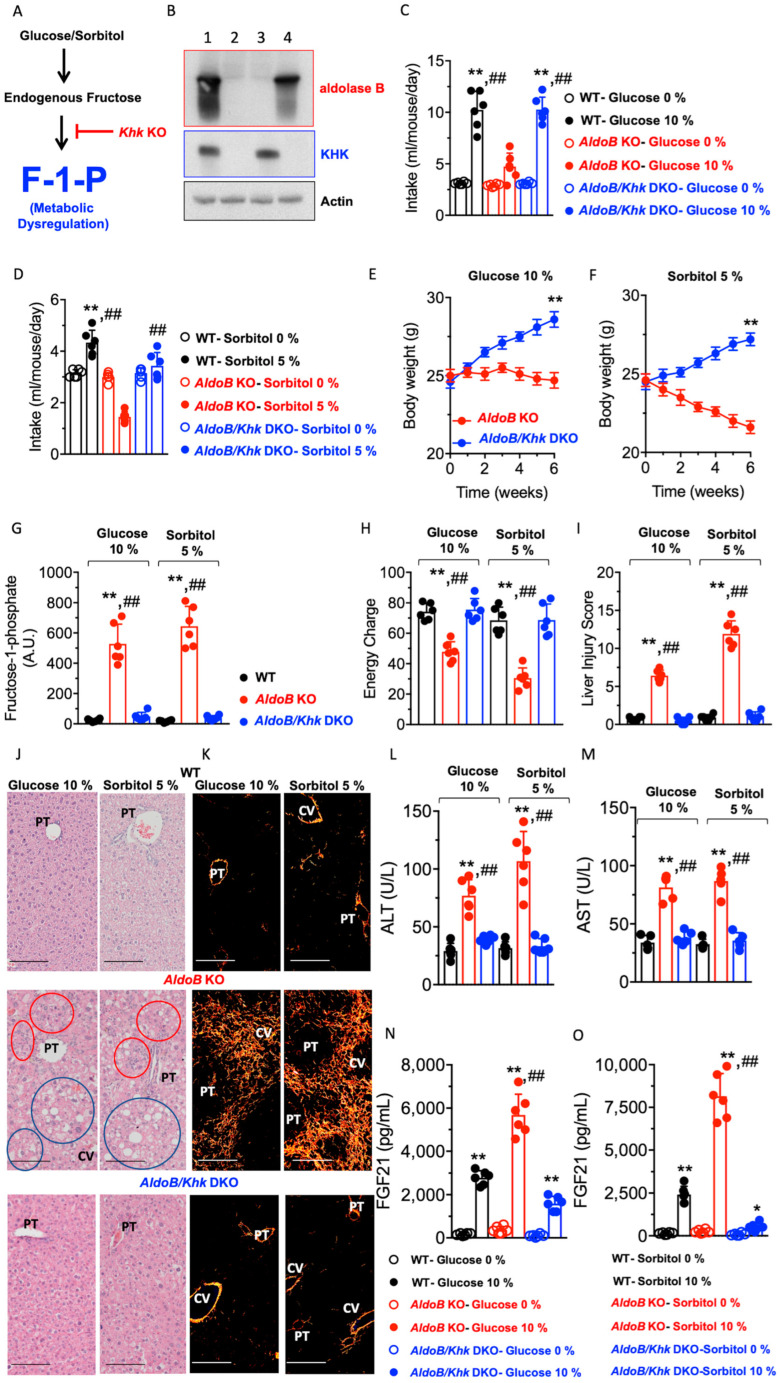
Deletion of KHK protects aldolase B KO mice against the deleterious effects of glucose and sorbitol. (**A**) Proposed schematic proposing that the deletion of *Khk* eliminates the accumulation of fructose-1-phosphate (F1P), thus protecting against metabolic dysregulation. (**B**) Representative Western blot from liver extracts for aldolase B, KHK and actin control in wild-type (lane 1), aldolase B and *Khk* double KO mice (*AldoB*/*Khk* DKO, lane 2), aldolase B KO mice (lane 3) and KHK KO mice (lane 4). (**C**) Average daily intake of 0 or 10% glucose in wild-type (WT, black), aldolase B KO (red) and *AldoB*/*Khk* DKO (blue) mice. (**D**) Average daily intake of 0 or 5% sorbitol in wild-type (WT, black), aldolase B KO (red) and *AldoB*/*Khk* DKO O (blue) mice. (**E**) Body weight gain in aldolase B KO and *AldoB*/*Khk* DKO mice consuming 10% glucose. (**F**) Body weight gain in aldolase B KO and *AldoB*/*Khk* DKO mice consuming 5% sorbitol. (**G**) Intrahepatic F1P levels in wild-type, aldolase B KO and *AldoB*/*Khk* DKO mice consuming 10% glucose or 5% sorbitol. (**H**) Liver energy charge in wild-type, aldolase B KO and *AldoB*/*Khk* DKO mice consuming 10% glucose or 5% sorbitol. (**I**) Liver injury score in wild-type, aldolase B KO and *AldoB*/*Khk* DKO mice consuming 10% glucose or 5% sorbitol. (**J**) Representative liver H&E images from wild-type, aldolase B KO and *AldoB*/*Khk* DKO mice consuming 10% glucose or 5% sorbitol (*w*/*v*) for 6 weeks. Blue circles denote macrosteatotic areas, and red circles indicate areas with ductal reaction and inflammation. Size Bar: 20 µM. PT: Portal triad. CV: Central vein. (**K**) Representative liver picrosirius red images under polarized light from wild-type, aldolase B KO and *AldoB*/*Khk* DKO mice consuming 10% glucose or 5% sorbitol (*w*/*v*) for 6 weeks. Size Bar: 20 µM. (**L**,**M**) Plasma ALT and AST levels in wild-type, aldolase B KO and AldoB/KHK DKO mice consuming 10% glucose or 5% sorbitol. (**N**,**O**) Plasma FGF21 in wild-type, aldolase B KO and *AldoB*/*Khk* DKO mice consuming 10% glucose or 5% sorbitol. The data in (**C**–**I**) and (**L**–**O**) were presented as the means ± SEM and analyzed by one-way ANOVA with Tukey post hoc analysis. For (**C**) and (**G**–**O**), * *p* < 0.05 ** *p* < 0.01 versus WT and ## *p* < 0.01 versus AldoB/KHK DKO. For (**E**,**F**), ** *p* < 0.01. *n* = 6 mice per group.

**Figure 5 nutrients-15-04376-f005:**
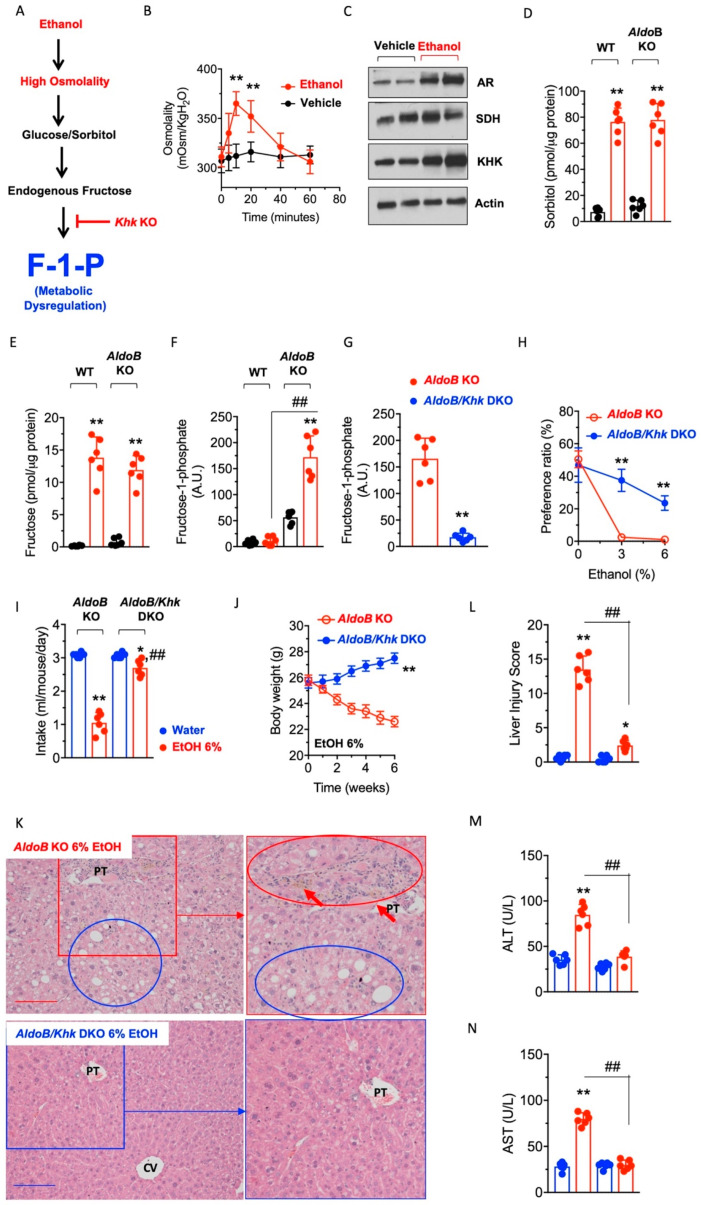
Effects of alcohol in the activation of the polyol pathway and metabolic dysfunction in aldolase B KO mice. (**A**) Schematic proposing that ethanol activates the polyol pathway in the liver via its hyperosmolar effects and that the deletion of *Khk* eliminates the accumulation of fructose-1-phosphate (F1P), thus protecting against metabolic dysregulation induced by ethanol. (**B**) Portal vein osmolality in aldolase B KO mice after receiving a 2.5 g/kg ethanol bolus. (**C**) Representative Western blot from liver extracts for aldose reductase (AR), sorbitol dehydrogenase (SDH), KHK and actin control in aldolase B KO mice consuming a 6% ethanol solution for 6 weeks. (**D**) Intrahepatic sorbitol levels in wild-type and aldolase B KO mice consuming 0 or 6% ethanol for 6 weeks. (**E**) Intrahepatic fructose levels in wild-type and aldolase B KO mice consuming 0 or 6% ethanol for 6 weeks. (**F**) Intrahepatic F1P levels in wild-type and aldolase B KO mice consuming 0 or 6% ethanol for 6 weeks. (**G**) Intrahepatic F1P levels in aldolase B KO and AldoB/KHK DKO mice consuming 0 or 6% ethanol for 6 weeks. (**H**) Two-bottle choice preference ratio for 0, 3 and 6% ethanol in aldolase B KO and *AldoB*/*Khk* DKO mice. (**I**) Average daily intake of 0 (water) or 6% ethanol in aldolase B KO and *AldoB*/*Khk* DKO mice on single bottle (no choice) for 6 weeks. (**J**) Body weight gain in aldolase B KO and *AldoB*/*Khk* DKO mice consuming 6% ethanol for 6 weeks. (**K**) Representative liver H&E images from aldolase B KO and *AldoB*/*Khk* DKO mice consuming 6% ethanol for 6 weeks. Blue circles denote macrosteatotic areas, and red circles indicate areas with ductal reaction and inflammation. Red arrows point to inflammatory foci. Size Bar: 20 µM. PT: Portal triad. CV: Central vein. (**L**) Liver injury score in aldolase B KO and *AldoB*/*Khk* DKO mice consuming 6% ethanol for 6 weeks. (**M**,**N**) Plasma ALT and AST levels in aldolase B KO and *AldoB*/*Khk* DKO mice consuming 6% ethanol for 6 weeks. The data were presented as the means ± SEM and analyzed by two-tail t test (**B**,**H**,**J**) or one-way ANOVA with Tukey post hoc analysis. For (**B**,**H**,**J**), ** *p* < 0.01. For (**D**–**F**), ** *p* < 0.01 versus respective strain vehicle (water) and ## *p* < 0.01. For (**I**–N) * *p* < 0.05 and ** *p* < 0.01 versus respective strain vehicle (water) and ## *p* < 0.01. *n* = 6 mice per group.

## Data Availability

Further information and requests for resources and reagents should be directed and will be fulfilled by the Lead Contact, Miguel A. Lanaspa (miguel.lanaspagarcia@cuanschutz.edu). Mouse lines generated in this study are available for any researcher upon reasonable request. This study did not generate unique datasets or code.

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
