# Peer review of "Endogenous Fructose Production and Metabolism Drive Metabolic Dysregulation and Liver Disease in Mice with Hereditary Fructose Intolerance"

_nutrients, 2023, doi:10.3390/nu15204376_

Round 1

Reviewer 1 Report

Andres-Hernando et al demonstrate the involvement of endogenous fructose in driving disease manifestations in hereditary fructose intolerance (HFI). Through the use of Aldob and Khk knockout mice, along with an aldolase reductase inhibitor, they clearly establish the role of polyol pathway in glucose- or ethanol-induced symptoms of HFI. The data presented in this study are well organized, and the conclusions draw are solid. However, there are a few minor points that could enhance the quality and insights of this manuscript.

1. As mentioned in the introduction, the tissue distribution of the enzymes involved in polyol pathway is not limited to a single tissue, such as the liver. They are also regulated under various stress conditions. Hence, the tissue expression levels of AR, aldolase B and KHK in each treatment used in this study should be investigated. Furthermore, discussing the tissue contributions of these finding and potential tissue crosstalk during glucose- or ethanol-induced symptoms in HFI could provide additional insights into the mechanisms underlying HFI.

2. It is important to consider the possibility that the uptake of glucose, sorbitol and ethanol may be influenced by central sensing mechanisms, rather than solely being a result of intolerance. The authors should not overlook this possibility and should thoroughly discuss it.

3. When presenting statistical information in multiple comparisons, it is crucial to clearly indicate the significance between which two groups are being compared in the figure, eg. Fig 1H-K.

4. The figure legend should provide specific information, including the age and sex of the mice, and the tissue examined, eg. Fig 2G, the comparison method used, sample size, and exact P values instead of providing a range.

5. Page 14 line 417-419, the description of “Low AMPK activation as a result of low free phosphate and ATP” is incorrect and needs to be revised.

6. All genes names should be written in Italics, and mouse genes should have the first letter capitalized.

7. On page, 3 line 119, there is an extra period that should be removed. 

Reviewer 2 Report

The study by Andres-Hernando et al. is an in vivo experimental study to evaluate the contribution of fructose endogenous production from glucose, sorbitol and ethanol by the aldolase reductase-sorbitol dehydrogenase (polyol) pathway. Although the in vivo studies are well-designed and the animal models used highly relevant, the rationale behind the working hypothesis should be further discussed in the manuscript. In fact, the authors state that subjects with HFI often switch to glucose-rich foods.  However, this is not the case, as HFI patients show a natural aversion to high glucose foods, similar to what is observed in here with the AldoB KO animals when given high glucose, sorbitol or ethanol. 

In line 64 of the Introduction the authors wrote that subjects with HFI tend to eat dextrose (glucose)-rich desserts. The authors must add references to support these findings as the common understanding is that HFI patients show aversion to glucose-rich foods.

In Figure 1, representative liver histology microphotographs of AldoB KO animals maintained on glucose 0%,1%, 5%, and 15% should be shown. Hepatic levels of F1P under these conditions is also relevant and should be quantified.

In all the animal models studied, aldolase reductase activity and the sorbitol dehydrogenase activity, that could be measured by commercially available kits, should be measured to confirm increased polyol pathway activity in AldoB KO animals on high glucose, sorbitol an ethanol. Or at least, WB of AR and SDH should be shown for all the animal models studied.

Lipogenic markers by WB analysis should be shown in the mouse models studied to confirm the hypothesis suggested by the authors in previous publications that increased DNL from endogenously synthetized fructose in ALDOB KO animals contributes to liver steatosis and injury.

The authors conclude that “In summary, our study provides evidence to help explain the  generalized, poor health  observed in subjects with HFI despite the avoidance of dietary fructose”. This sentence should be rewritten; as this study has been made in mice under the hypothesis that HFI patients consume a high glucose diet, no conclusions but only hypothesis can be drawn from these results relating to patients with HFI.

Round 2

Reviewer 2 Report

The authors have correctly addressed the raised concerns. Therefore, according to this Reviewer,  this manuscript is suitable for publication.